# Contribution of Oxidative Stress (OS) in Calcific Aortic Valve Disease (CAVD): From Pathophysiology to Therapeutic Targets

**DOI:** 10.3390/cells11172663

**Published:** 2022-08-27

**Authors:** Daniela Maria Tanase, Emilia Valasciuc, Evelina Maria Gosav, Mariana Floria, Claudia Florida Costea, Nicoleta Dima, Ionut Tudorancea, Minela Aida Maranduca, Ionela Lacramioara Serban

**Affiliations:** 1Department of Internal Medicine, Grigore T. Popa University of Medicine and Pharmacy, 700115 Iasi, Romania; 2Internal Medicine Clinic, St. Spiridon County Clinical Emergency Hospital Iasi, 700111 Iasi, Romania; 3Department of Ophthalmology, Faculty of Medicine, Grigore T. Popa University of Medicine and Pharmacy, 700115 Iasi, Romania; 42nd Ophthalmology Clinic, Prof. Dr. Nicolae Oblu Emergency Clinical Hospital, 700309 Iasi, Romania; 5Department of Morpho-Functional Sciences II, Discipline of Physiology, Grigore T. Popa University of Medicine and Pharmacy, 700115 Iasi, Romania; 6Cardiology Clinic St. Spiridon County Clinical Emergency Hospital, 700111 Iasi, Romania

**Keywords:** oxidative stress, calcific aortic valve disease, bicuspid aortic valve, valvular interstitial cells, antioxidants, therapeutic targets

## Abstract

Calcific aortic valve disease (CAVD) is a major cause of cardiovascular mortality and morbidity, with increased prevalence and incidence. The underlying mechanisms behind CAVD are complex, and are mainly illustrated by inflammation, mechanical stress (which induces prolonged aortic valve endothelial dysfunction), increased oxidative stress (OS) (which trigger fibrosis), and calcification of valve leaflets. To date, besides aortic valve replacement, there are no specific pharmacological treatments for CAVD. In this review, we describe the mechanisms behind aortic valvular disease, the involvement of OS as a fundamental element in disease progression with predilection in AS, and its two most frequent etiologies (calcific aortic valve disease and bicuspid aortic valve); moreover, we highlight the potential of OS as a future therapeutic target.

## 1. Introduction

Valvular heart disease (VHD) is currently a major contributor to the loss of physical functioning, quality of life, and longevity [1]. Between 1990 and 2017, the global burden of VHDs showed a wide range of variations; over this time, period rheumatic heart disease, considered the leading cause of VHD in developing countries, substantially decreased. In this matter, The Euro Heart Survey (EHS) noted a predominance of functional and degenerative valvular diseases in high-income countries, where degenerative aortic valve disease and degenerative mitral valve disease represented 70% and 46% of VHD cases [1,2,3,4]. Additionally, the elderly population (≥60 years) is expected to grow from 962.3 million in 2017 to 2080.5 million in 2050, which will lead to a substantial increase in the incidence of non-rheumatic VHD (NRVHD), and its most frequent subtype—non-rheumatic calcific aortic valve disease (NRCAVD) [1,2].

Aortic stenosis (AS) is the most common type of valvular heart disease [5]; its prevalence varies between 2% and 7% in patients over 65 years old and it is estimated that by the year 2030, 4.5 million patients will be diagnosed with this VHD [6]. The most common etiology of aortic stenosis is calcific disease; however, the most common congenital valve defect causing AS is the bicuspid aortic valve (BAV) [7]. Calcific aortic valve disease (CAVD) is no longer considered a simple passive calcium depositing process with consequent fibrosis and valve leaflets thickening, it is also an active cellular process that causes endothelial dysfunction throughout oxidative stress (OS) and inflammatory activation [8,9]. BAV is characterized by the abnormal cusp development where two of the three cusps fuse into a single cusp resulting in abnormal leaflet architecture; it has a prevalence of 0.5–2% in the general population and is three times more prevalent in males than in females [10,11]. One of the most frequent complications of this congenital malformation is BAV-associated aortopathy (BAVAA), which predisposes patients to the risk of developing thoracic aortic aneurysm, aortic dissection in the ascending aorta, and is linked to a faster aortic valve degeneration [10,12]. In both aortic diseases, valvular interstitial cells initiate and maintain mineralization, calcification, and fibrosis; all of these processes are supported by mechanical and biochemical pathways that include lipid deposition, inflammation, and OS [13,14].

In this review, we describe the mechanisms behind aortic valvular disease and the involvement of oxidative stress as a key element in disease progression, with predilection in AS and its two most frequent etiologies (calcific aortic valve disease and BAV); moreover, we highlight the potential of OS as a future therapeutic target.

## 2. Immunopathogenesis behind Aortic Disease

Aortic valve disease is a growing health condition that has a considerable impact on quality of life. The lack of effective strategies to prevent aortic valve degeneration and progression into calcific aortic valve stenosis (CAVS), besides the surgical option, reinforces the necessity of comprehending the immunopathogenic mechanism in order to develop targeted therapies [15,16].

Genetic susceptibility, endothelial shear stress, chronic inflammation, oxidative stress, lipid deposition, and valve calcification are all contributors to the progression of fibrocalcific remodeling of the aortic valve. The presence of overlapping disease development risk variables and a link between the severity of CAVD and coronary calcification suggest a shared disease pathway, at least in the early stages between CAVD and atherosclerosis [8,17]. As seen in atherosclerosis, endothelial dysfunction secondary to mechanical stress is the initiating event in CAVD. Damaging the structure of the endothelium enables lipids, primarily low-density lipoproteins (LDL) and lipoprotein (a) (Lp(a)) to penetrate the aortic leaflets, and can also trigger the migration of inflammatory cells within [17,18]. Plaque-like lesions consisting of LDL, Lp(a), and inflammatory cells form in the subendothelial region contributing to the production of reactive oxygen species (ROS) that promotes the oxidation of LDL and vascular dysfunction [17,19,20]. The persistent inflammatory process enables valvular interstitial cells (VICs) to attain an osteogenic phenotype resulting in the production of microcalcifications and leaflet mineralization [7,11]. The end stages of CAVD and atherosclerosis differ in the main cell types involved. Whilst in atherosclerosis, smooth muscle cells are known to be the major effectors in chronic inflammation, in CAVD, the pathophysiological process is enhanced by the addition of fibroblasts [17,21,22].

Given the various treatments that have been proven to delay the progression of atherosclerosis, none are effective at preventing aortic stenosis [7]. By stimulating inflammation through a deficiency of antioxidant protection (superoxide dismutase, heme oxygenase-1, glutathione peroxidases) or excessive formation of ROS, OS draws attention as one of the key mechanisms in the early stages of the disease [23,24]. Standard cardiovascular medications evaluated in clinical trials failed to slow disease progression or reduce negative outcomes. More research into alternative therapeutic options that target oxidative stress in VHD is needed [25].

### 2.1. Mechanical Stress and Endothelial Dysfunction

Endothelial dysfunction is recognized as a primary phase in the genesis of vascular disease and one of the most promising domains for the development of new therapeutic targets [26]. Mechanical stress, lipid infiltration, OS, inflammation, and the fibro-calcific response are all examples of interrelated events that trigger endothelial dysfunction, translated as the loss of its barrier properties [18,27]. The repeated opening and closing of aortic valvular cusps during the cardiac cycle is a continuous source of mechanical stress, which contributes directly to compromising the structural stability of the endothelial layer [5,18,28].

The aortic side of the valve is usually the area of choice for the above-mentioned pathological modifications, especially the noncoronary leaflet [28]. During systole, the blood is forcibly ejected past the ventricle leaving the aortic surface of the valve exposed during the diastole to a higher velocity gradient blood flow and oscillatory shear stress [29,30]. The bicuspid aortic valve also depicts the key role of hemodynamic stress through the high prevalence, accelerated initiation, and progression of CAVD [27,28], mainly at the site of the fused BAV leaflet [11]. In order to properly maintain endothelial homeostasis on both valvular and vascular endothelial cells (VECs), laminar blood flow is a mandatory condition. Fulfilling this requirement, the endothelial relaxing factors (mainly the nitric oxide signaling pathway) are stimulated while the expression of adhesion molecules (vascular cell adhesion molecule 1 (VCAM-1), intercellular adhesion molecule 1 (ICAM-1), E-selectin), and vasoconstrictors (endothelin-1, angiotensin II, thromboxane A2) are suppressed [27,31]. The most common local laminar flow disturbances are inflicted by atherosclerosis, arterial stiffening, and remodeling [32], arterial hypertension, or valvular disorders [27,31]. Valvular endothelial cell changes relate to shear stress by aligning perpendicular to the flow direction opposed to vascular endothelial cells, which align parallel to the flow [29,30].

Ontological and transcriptional profiles suggest that valve endothelial cells (VECs) have the particularity to display side-specific heterogeneity being innate suppressors of calcification with antioxidative and anti-inflammatory phenotypes [30,33,34,35]. VECs have the ability to not only detect alterations in sheer stress but also modulate these signals by regulating valvular interstitial cell (VIC) functions, preventing them from differentiating into osteoblast-like cells [11]. The process is very similar to the endothelial to mesenchymal transition (EndMT) seen throughout valve development and formation [21,36].

Apart from the classic wear and tear pathogeny, biomechanical factors are now recognized as driving forces in the development of CAVD [37,38]. For example, genes, such as the KLF2/PLPP3 axis, inhibitor of DNA binding-1 (ID1), heme oxygenase 1 (HMOX1), and nitric oxide synthase 3 (NOS3) may play hemodynamic-related roles in the development of CAVD by regulating other procalcification genes, such as nuclear factor kappa-light-chain-enhancer of activated B cells (NF-κB) [39], by promoting endothelial repair [40] and by protecting against OS and atherosclerosis [37].

The association between hemodynamics and endothelial oxidative stress in the genesis of CAVD is not yet completely defined and lacks sufficient research [37].

### 2.2. Lipid Deposition and Oxidative Stress in SA and BAV

Nowadays, western dietary patterns (consisting of a high intake of saturated fatty acids, processed foods, and low intakes of unsaturated fatty acids, dietary fiber, and micronutrients) combined with sedentary lifestyles and increased life expectancies have, as a consequence, contributed to the increase in noncommunicable diseases (NCDs), such as cardiovascular diseases, diabetes, and cancers [41,42]. Vascular OS and inflammation are clearly distinguished mechanisms underlying endothelial dysfunction, which lowers the bioavailability of the key vasodilator molecule nitric oxide (NO) and promotes inflammatory cell recruitment, lipid peroxidation, and aortic valve remodeling [32,43].

Uncoupled nitric oxide synthases (NOS), nicotinamide adenine dinucleotide phosphate oxidases (NOXs), and mitochondrial ROS (mitoROS) are the main sources of ROS in the cardiovascular system [44]; they maintain endothelium homeostasis and smooth muscle cell contraction [45]. Since its first definition in 1985, OS has evolved into a global concept featuring an imbalance between ROS formation and antioxidant response [46]. Interactions between the key ROS generating systems (NOXs, xanthine oxidase, NOS, myeloperoxidase, lipoxygenases) and the major antioxidant systems (catalase, superoxide dismutase (SOD), glutathione peroxidase, glutathione S transferases (GST), α-tocopherol, ascorbic acid, reduced glutathione, and protein thiols) has generated a balanced redox state [47,48,49]. In recent years, the nuclear factor erythroid 2-related factor 2 (Nrf2) has gained attention as an important mediator in the homeostasis of ROS via the Keap1/Nrf2 axis [50,51,52,53].

Although oxidative stress and atherosclerosis are prominent traits in CAVD, statins used as successful monotherapies in atherosclerosis have not been proven effective in CAVD treatment amidst the significant evidence that LDL contributes to disease progression [6,18,20,54]. The depositions of oxidized LDL and LP(a) molecules in the subendothelial layer sustain chronic inflammation at the expense of proinflammatory cytokines and fibrocalcific remodeling by promoting osteogenic differentiation of VICs (via toll-like receptors 2 and 4) and the upregulation of bone morphogenetic protein 2 (BMP-2) (via extracellular signal-regulated kinase (ERK)1/2 and NF-κB) [7,14,21,27,55].

From the four NOS subtypes, endothelial nitric oxide synthase (eNOS) plays a key role in NO modulation being diminished under elevated oxidized LDL (oxLDL) exposure. eNOS uncoupling [31,37,56] and tetrahydrobiopterin (BH4) depletion (NOS co-factor) are two major traits in CAVD that lead to ROS cell injury [57,58]. In addition to eNOS uncoupling, several more pathways, such as mechanical stress [34,37], transforming growth factor beta 1 (TGF-β1) stimulation [36], and S-glutathionylation protein are involved in NO impaired production [59]. BAV aortopathy studies on OS reiterate the role of NO in endothelial homeostasis [60,61,62].

At the ROS imbalance, besides eNOS uncoupling in BAV aortopathy, the ineffective antioxidant protection is demonstrated by the inadequate SOD activity in the ascending aorta [11,63,64,65], whose expression is also dependent on the valve morphotypes [10,12,66,67]. Alongside NOS, NOXs are also part of the ROS milieu seen in CAVD. NOX2 and NOX4 are the main members of the NOX assembly (NOX1 to 5 and DUOX1 and 2) in the vascular system that are required for LDL oxidation, as shown in experimental mice studies [68]. However, the exact role of every NOX in CAVD is not very well defined and further studies are needed. The significance of NOX2 in CAVD has been illustrated in hypercholesterolemia-induced mice who received western-type diets [69], in a rabbit model fed cholesterol-enriched chow and vitamin D [70], and in human aortic tissue [44,71]. From the standpoint of mitoROS, oxidative stress generated through this pathway is observed in the VIC initiation of calcific degeneration [72].

In conjunction with the direct mechanism that NOX exerts, it is also intricated in the crosstalk between OS and inflammation through the cytokine release, controlling the nod-like receptor protein 3 (NLRP3) inflammasome [25,73,74,75,76]. Additionally, ROS assists various inflammation pathways via the redox modulation of inflammation mediators (damage-associated molecular patterns (DAMP)s, high mobility group box 1 (HMGB1), and S100 proteins), transcription factors (Nrf2, NF-κB, HIF- 1α, and AP-1), and by the formation of redox-dependent protein complexes (Nrf2-Keap1) [25,49]. Semicarbazide-sensitive amine oxidase (SSAO)/vascular adhesion protein-1 (VAP-1) has been identified as an additional injurious component of ROS in CAVD and has been linked to disease activity in several research studies [16,44,77]. ROS components cannot be viewed as separate effectors because they are part of an intricate crosstalk process supporting self-propagation and amplification of oxidative stress [25,49].

### 2.3. Inflammation Mechanism

Apart from being present from the onset of CAVD, inflammation orchestrates all subsequent events that lead to aortic valve remodeling [78]. In summary, as part of the initial innate immune response, activated VICs from prior exposure to mechanical stress and cytotoxic oxLDL prompt the recruitment and infiltration of macrophages, T cells, and mast cells via toll-like receptors 2, 4 (TLRs-2,4) and the NF-kB pathway [6,14,15,17,21,37].

At the same time, extracellular matrix remodeling and osteogenic gene expression take place under the effects of profibrotic cytokines (IL-2, IL-1β, TNF-α, IL-6) and proteolytic enzymes (collagenase-1/MMP-1, collagenase-3/MMP-13, gelatinase-A/MMP-2, and gelatinase-B/MMP-9) [20,34,79]. For the induction of vascular inflammation, the activation of the NF-kB pathway is pivotal and can be achieved either by the canonical (mediated by tumor necrosis factor-alpha (TNF-α), interleukin-1β (IL-1β), angiotensin II-induced ROS), or by the non-canonical pathway (mediated by vasoactive peptides, ox-LDL, activated CD40 receptor, B cell-activating factor (BAFF), lymphotoxin b (LTb) monocyte-released cytokines, and advanced glycation end-products) [40,80,81].

Amongst the interleukins, the IL-6 through NF-kB/ interleukin-6/bone morphogenic protein (BMP) signaling pathway [14,82] expression of the receptor activator of the NF-kB ligand superfamily member 11 (RANKL) [7] and IL-1 through enhancing the expression of matrix metalloproteinases (MMPs) [8,17,73] sustain inflammation involvement in aortic valve remodeling and calcification. Amidst the cytokines, TNF-α is of particular importance as a direct inductor of the NF-κB canonical pathway, further mediating biomineralization of the valve [27] and activation of ERK1/2, c-Jun N-terminal kinase (JNK), and p38/MAPK [55]. Being part of a complex, intricate, and self-amplifying mechanism, the roles of IL-1, IL-6, and TNF-α cannot be fully divided, together with IL-18 and IL-32 promoting the secretion of IL-8, MCP-1/CCL2, and adhesion molecules [5,17,27,83].

As a peculiarity in BAV, enhanced densities of macrophages [11] that produce IL-12 and Th17 activation have been observed, as well as changes in the proteomics of canonical pathways characterized by significant upregulation of α-1-antitrypsin (AAT), α-1-antichymotrypsin (ACT), and α-1-acid glycoprotein 2 (AGP) [64]. In addition to macrophages and inflammatory cells, a cluster of CD4+ and CD8+/CD28− T cells are present in early valve lesions adjacent to calcific loci [82,84].

As far as the adaptive immune response in aortic stenosis is concerned, this remains a topic with considerable potential for further research, especially with regard to intracellular pathogens [17] or epitopes [82].

### 2.4. Extracellular Matrix Remodeling and Biomineralization

A starting point in extracellular matrix fibrosis is the differentiation of VICs into active myofibroblasts via the Wnt3a/β-catenin pathway and TGF-β1-SMAD2/3 activation that upregulates the SMC markers α-SMA and SM22αn [34,79,85]. α-SMA levels are also increased by the renin-angiotensin system (RAS), which alongside BMP-2, ALP, MSX2, RANK, and RANKL subsequently favor the VIC osteogenic phenotype [31,86]. The positive pro-fibrotic feedback loop created by TGF-β1 is further sustained by integrins and adhesion kinases being mediated by phosphoinositide 3-kinase (PI3K)/Akt, the Ras homolog gene family, member A and its associated kinase (RhoA/ROCK), ERK1/2, JNK, p38/MAPK [55], and NF-κB [24,37,72,87].

Over the course of CAVD progression under the influence of periostin, TNF-α, IL-1β, TGF-β1, RANKL, and excessive strain matrix metalloproteinases (MMPs) (especially MMP-1/9/12) and their tissue inhibitors (TIMPs) are secreted by ECs, myofibroblasts, macrophages, and T cells modulating collagen turnover in fibrotic aortic leaflets [6,19,37,88,89,90]. In addition to MMPs and TIMPs, cathepsins S/K/V [10,11,66,84] and elastin-degrading proteases enhance the remodeling and degradation of the vascular extracellular matrix (VECM) [89,91]. Driven by hypoxia in thickened leaflets, neoangiogenesis occurs in areas of calcification surrounded by intensive inflammation and is accompanied by circulating osteogenic progenitor cells [7,8,11].

Degradation and fibrosis of the extracellular matrix serve as the foundations for the two subsequent calcification processes that follow. Dystrophic calcification is enabled by the precipitation of calcium and phosphates in potent calcification nuclei, such as degraded collagen and elastin fibers in the absence of the active participation of osteoblasts [77]. In the definitive stages of CAVD evolution, dystrophic calcification is completed by biomineralization. The process is governed by VICs differentiated into osteoblasts under the transcription factor RUNX2, acquiring the expressions of ALP, BMP-2, BMP-4, osteopontin (SPP1), osteocalcin (BGLAP), Sry-related HMG box-9 (SOX9), and bone sialoprotein [6,34,53]. Under hypercholesterolemia and oxidative stress, endothelial valvular cells secrete Wnt3 that binds to receptors LRP5 and Frizzled on the VIC surface [34,58,82], further inducing the RUNX2 expression via the Wnt pathway [7]. The Notch signaling pathway, consisting of four receptors (Notch1/2/3/4), ligands, Jagged-1/2, and delta-like proteins (DLL1/3/4), is a physiological neutralizer of the Wnt pathway, upholding indirect regulatory effects on RUNX2 [20], being in close conjunction with NF-κB and TLR4 activation in calcific areas of aortic leaflets [60,82].

The osteoprotegerin (OPG)/RANKL/RANK axis represents an alternative route that controls osteoclastogenesis through its receptor RANK and VIC osteoblastic transformation under RANKL overly-expressed by T cells and macrophages in stenotic valves [21,84]. Increased RANKL in atherosclerotic aortic valves interferes with OPG, the antagonist of RANKL that precludes osteoblast maturation [5,17]. In osteogenic transformation, bone morphogenetic proteins (BMPs) in conjunction with the SMAD pathway, RUNX2, Osterix, MSX2 [57], and DIX5/6 uphold significant roles for osteoblasts and chondrocytes [6,20]. As far as VIC osteoblastic transformation is concerned, BMP-2 and BMP-4 play key roles in aortic stenosis alongside ERK1/2, SMAD, ALP, RUNX2, and osteopontin [21,22,28,34,80,84].

In summary, the interplay of oxidative stress, endothelial dysfunction, and inflammation alter valvular homeostasis expressed as a damaged and hyper-vascularized extracellular matrix containing VICs differentiated into active myofibroblasts (Figure 1). The anarchic organizational pattern of collagen fibers promotes calcium and phosphate deposition, and VIC osteogenic differentiation, conclusively leading to valve ossification, and ending the complex active chain reaction of biomineralization [27].

## 3. Therapeutic Targets of Oxidative Stress

CAVD is extremely common and a rapidly advancing disease with no effective drug therapies at present. Surgical aortic valve replacement (SAVR) is the last resort in severe cases;the window of opportunity being limited for elderly frail patients as they represent the majority of concerned patients and, thus, it is not the most desirable option [4,7,92]. In the last decades, transcatheter aortic valve replacement (TAVR) gained popularity as a more tolerable procedure with promising outcomes [48,93,94,95].

Several therapies that were attempted, including statins, antihypertensives, and medications that target the phospho-calcic pathways, were unable to arrest the naturally occurring progression of CAVD [14,20,96]. As the research into the pathology of calcific aortic stenosis progressed and revealed the multifactorial processes and cellular mechanisms, the demand for targeted therapeutics effectively preventing or delaying the progression of CAVD has emerged [17,20,97,98,99].

Regarding oxidative stress as a unifying mechanism involved in the initiation and propagation phases of aortic stenosis, counteracting it may represent a major future in targeted therapy. Thus, Table 1 provides an overview of potential strategies for reducing the interplay of oxidative stress in CAVD.

### 3.1. Targeting Lipid Oxidation

Since atherosclerosis and CAVD share similar risk factors and pathways in the initial pathological stages, targeting LDL to further prevent oxLDL formation was the first subject of interest in regard to medical therapies for aortic stenosis.

Four large-scale randomized controlled clinical trials (RCT), registering a total of 2388 patients, showed no significant improvements in CAVD progression after LDL cholesterol-lowering treatment [18,54,100], with only statins (rosuvastatin, atorvastatin) in ASTRONOMER (Aortic Stenosis Progression Observation: Measuring Effects of Rosuvastatin) (NCT00800800) [101], RAAVE (Rosuvastatin Affecting Aortic Valve Endothelium) (NCT00114491) [102], and SALTIRE II (Scottish Aortic Stenosis and Lipid Lowering Trial, Impact on Regression) (NCT02132026) trials [103], or statins in association with fibrates (simvastatin/ezetimibe) in the SEAS (the Simvastatin and Ezetimibe in Aortic Stenosis) (NCT00092677) trial [104]. The conflicting outcomes were proven to be the result of increased levels of Lp(a) [101,105,106,107]. Hence, studies started to focus on the correlation between Lp(a) and CAVD [14,108], confirming that Lp(a) levels are associated with an increased risk of surgical replacement supplemented by complications, hemodynamic alterations [109,110,111,112], and cardiac death [113,114], reinforcing the premise that lowering Lp(a) may be a future effective treatment.

Proprotein-converting enzyme subtilisin/kexin type 9 (PCSK9) inhibitors can simultaneously decrease the level of LDL-C via LDL-R internalization and Lp(a) synthesis [115,116]. The FOURIER (Further Cardiovascular Outcomes Research With PCSK9 Inhibition in Subjects With Elevated Risk) (NCT01764633) [115,117] and the GLAGOV (Global Assessment of Plaque Regression with a PCSK9 Antibody as Measured by Intravascular Ultrasound) (NCT01813422) studies [118] reinforced the important roles of PCSK9 inhibitors in CAVD treatment.

Niacin (nicotinic acid) also raises awareness by reducing Lp(a), LDL-C, and triglycerides plasma levels and by increasing HDL-C [119]. There are currently only two trials evaluating niacin (the Early Aortic Valve Lipoprotein (a) Lowering (EAVaLL; NCT02109614)) and PCSK9 inhibitor (South Korean clinical trial (NCT03051360)), supporting the necessity of additional studies on this matter [6,14,34,108].

Presently, there is a paucity of data from RCTs to endorse the notion that lowering Lp(a) levels alone can improve the CAVD outcome without taking into account other lipid risk factors. New therapies consisting of antisense oligonucleotide against apolipoprotein(a), which consistently lowers plasma Lp(a) levels, such as IONIS-APO(a)-Rx, IONIS-APO(a)-LRx [120], and hepatocyte-directed antisense oligonucleotide AKCEA-APO(a)-LRx (TQJ230) have recently emerged [106]. Additionally, E06, a natural antibody that prevents macrophages from absorbing oxLDL, demonstrated a 49% reduction in the AV gradient and a lower overall calcium load in E06 mice [121].

Alternative therapies that inhibit LDL oxidation and that are not yet proven to be directly correlated to CAVD may be represented by probucol [24], sitagliptin, and alogliptin [34,122]. Moreover, aegeline, a compound anti-lectin-like oxidized-LDL receptor-1 (LOX-1) [77], as well as capsaicin [123], kefir peptides [73], and p38 MAPK inhibitors dilmapimod (SB681323, GlaxoSmithKline, London, UK) and losmapimod (GW856553, GlaxoSmithKline, London, UK) [55], are currently under investigation.

### 3.2. Antioxidants from Natural Compounds to Targeted Experimental Therapies

A general trait observed in calcified aortic valves is the general reduction of the major antioxidant systems and excessive ROS accumulation; hence, mechanisms approaching these imbalances were sought as novel therapeutic strategies [44,124,125]. Most natural antioxidants are obtained from herbal sources, as will be described in the following paragraphs.

#### 3.2.1. Phenolic Acids, Flavonoids, Anthocyanins, Lignans, and Stilbenes

The most known polyphenolic representative is likely resveratrol (3,5,4′-trihydroxytrans-stilbene), which is found in berries and grapes [81]. The major antioxidant effects are mediated by SIRT1, Nrf2, and the ER (estrogen receptor) reducing arterial calcification by downregulating RUNX2, osteocalcin, and ALP [83,126,127,128]. In addition to having an anti-inflammatory impact, precursors of resveratrol (palmitoylethanolamide, polydatin) significantly reduced the expression of VCAM, ICAM-1, TNF- α, IL-1, alleviating vascular injury [19,49,129,130,131,132].

Curcumin, historically used as a spice, and the main component of Curcuma longa (turmeric), reduces oxidative stress by inhibiting osteogenic VIC differentiation across NF-κB, protein kinase B (PKB/AKT), and extracellular signal-regulated kinase (ERK) routes, and modulates the formation of ox-LDL [133,134,135,136,137] while reducing the expression of pro-atherogenic cytokines [49,138].

Cardamonin, primarily available in Ginkgo biloba and Amomum subulatum, exhibited strong anticalcific effects by suppressing the activation of p-AKT, p-ERK1/2, as well as the expressions of NLRP3 inflammasome and IL-1 [139,140,141].

Improved endothelial dysfunction and anti-proliferative impacts on smooth muscle cells were exemplified by ellagic acid, naturally present in nuts and red berries [142,143], and gallic acid that further decreased the expressions of the calcification markers RUNX2, BMP2, osteocalcin, and MSX2 [87,144]. Anticalcific and anti-inflammatory properties by interfering with the PI3K-AKT, ERK1/2, and NF-κB/NLRP3 inflammasome pathways are also displayed by caffeic acid phenethyl ester (CAPE), found in honeybee propolis and conifer barks [145,146].

Flavonoids are potent calcification inhibitors interfering with PI3K-Akt, mTOR, NF-kB, and ROS/TLR4 via nobiletin found in citrus fruits [147,148,149], and quercetin, which additionally modulates the iNOS/p38 MAPK pathway [49,150,151,152,153].

Further anti-inflammatory, anti-calcific, and ROS-downregulating properties were observed in anthocyanins [26,154], phytoestrogen puerarin [155,156,157,158], diosgenin [159,160,161], and 10-dehydrogingerdione (10-DHGD) [162].

#### 3.2.2. Vitamins

Inhibitory effects on lipid oxidation, cardiovascular inflammation, vascular calcification, and the improvement of vascular endothelial dysfunction are depicted by vitamins C and E (α-tocopherol), which have been extensively researched in cardiovascular preventive medicine [19,163,164]. Several observational studies have revealed an unfavorable connection between the nutrient intake of antioxidants and cardiovascular morbidity and mortality, despite the supporting idea that antioxidant vitamins help reduce oxidative stress and vascular illnesses [165].

The cardiovascular morbidity and mortality rates in the placebo and antioxidant groups were found to be identical in meta-analyses of RCTs on the benefits of vitamins B, C, E, and S, or b-carotene [166,167]. The unsatisfactory results can be explained by a number of factors, such as the lack of standard dosages and the follow-up periods of insufficient lengths, but further research is needed on the so-called antioxidant paradox in therapeutics [45].

#### 3.2.3. Other Antioxidant Compounds

Fucoxanthin (found in brown seaweed and micro and macro algae) [87,168], Andrographolide extracted from a Chinese medicinal herb (Andrographis paniculata) [169,170,171], and Apocynin found in Apocynum cannabinum and Picrorhiza kurroa [172,173] have potent antioxidant properties and attenuated calcification by interfering with the AKT, NF-κB, and ERK1/2 signaling pathways. Additionally, apocynin and celastrol are isolated from the Tripterygium wilfordii Hook F (TWHF) plant, also known as Thunder God Vine lowered NOX expression [70,174,175].

Compounds found in Hengshun aromatic vinegar (HSAV), also from the Asian cuisine, display a myriad of antioxidants, such as polyphenols, flavonoids, and melanosine, enhancing SIRT1, Nrf2, and NQO1 expressions [176], alongside capsaicin, proving antioxidant efficacy, which may be considered for future studies [123].

Recently, the pineal hormone melatonin showed antioxidant effects in atherosclerotic lesions excising NLRP3 inflammasome activation [49,177]. In patients with underlying pathologies that additionally amplify ROS production, such as rheumatic valvular heart disease, or those undergoing cardiopulmonary bypass or diabetes, L-carnitine and glycine represent promising potential in targeting NF-κB, Nrf2 [178], and the RAGE-NOX-NF-κB signaling pathway [75]. Studies have also been conducted on ascending aortic aneurysms and patients with Marfan syndrome. Thus far, Hibiscus sabdariffa Linne (HSL), which is rich in polyphenols, anthocyanin, flavonoids, and L-ascorbic acid [19], and the inhibitor of the kappa B kinase epsilon (IKKε) inhibitor, amlexanox [179], are potential antioxidants meriting further research.

#### 3.2.4. Targeting mitoROS and SOD

One of the most important systems involved in maintaining redox balance is SOD, which is downregulated in aortic stenosis [24]. In an attempt to promote SOD upregulation, polyethylene glycol-SOD (peg-SOD) and other SOD-mimetics consisting of cerium oxide nanoparticle (CNPs) rods and sphere-shapes [23] were utilized [180,181]. With the application of several aortic VICs, controversial outcomes were observed in terms of the calcification process [70], raising the question as to whether this therapeutic method depends on the dosage or a specific CAVD model. A redox-active MnSOD mimetic, manganese compound MnTnBuOE-2-PyP5 (MnBuOE) reduced aortic valve remodeling by targeting collagen buildup [182].

Research on mitoROS is somewhat underdeveloped with only one compound being mentioned. Mitoquinone (MitoQ), a derivative of ubiquinone, downregulated OS through the Nrf2/Keap1 pathway [52], with further antifibrotic properties obtained by TGF-β1-NOX4-ROS axis suppression [85].

### 3.3. NO Bioavailability and NOX Inhibition

Increasing NO bioavailability was proposed as an alternative to reduce oxidative stress relying on the concomitant Notch1 regulation to also target osteogenic VICs differentiation [183]. From the aforementioned potential therapies, statins illustrate pleiotropic NO increased levels but their failure in aortic stenosis therapy mandates the use of alternative approaches [34].

These included treatment with NO donors application (DETA-NONOate) [184], sodium nitroprusside (SNP) [185], NO precursor L-arginine [186], and co-treatment with BH4, which mitigates eNOS uncoupling, showing modest results in attenuating osteogenic differentiation and endothelial dysfunction [15,187,188]. Consecrated therapies, such as angiotensin-converting enzyme (ACE) inhibitors [96], nebivolol [45], inhibitors of dipeptidyl peptidase 4 (DPP4), such as sitagliptin and anagliptin [89,122,189,190], and empagliflozin, a sodium–glucose cotransporter-2 (SGLT2) inhibitor [25], also interfere and prevent eNOS uncoupling and suppress superoxide production. Currently, there are no RCTs testing the aforementioned theories, lending credence that NO bioavailability modulators might be a cutting-edge therapy in CAVD in the future.

In the hope of finding a more targeted approach, NOX-specific inhibitors were explored. Apocynin and celastrol from the antioxidant category demonstrated their potential as NOX2-selective inhibitors in addition to other non-specific antioxidant activities described in previous paragraphs. Apocynin significantly prevents the phosphorylation of the NOX2 cytosolic component [172], and celastrol demonstrates a general favorable effect in terms of calcium deposition and AV fibrotic remodeling by blocking the NOX2-driven GSK3β/β-catenin pathway [70,191].

Antidiabetic therapies, such as sitagliptin, showed pleiotropic effects in NOX by additionally inhibiting the cross-talk between RAGE/NF-κB with implications in calcific remodeling [21,89,122]. By reducing TNF-α, Il-6, iNOS, and NOX2 expression, the GLP-1 analog liraglutide also reduced vascular dysfunction, aortic inflammation, and oxidative stress in mice with polymicrobial sepsis [192]. Given that numerous other anti-inflammatory properties of pioglitazone were documented, such as improving TNF and IL-6 levels and cusp mobility, it is unclear as to the specific role it plays in NOX2 inhibition [69,193]. Triazolopyrimidines, such as VAS2870 and VAS3947 [194,195,196], and pyrazolopyridine derivatives, such as GKT136901/137831 [197], were developed under the idea of creating substances that specifically block NOX enzymes, although more research is needed to determine their safety and precise effects in CAVD [45,56,63]. As seen, efforts are being made in finding new therapeutic approaches via OS in CAVD, with potential for future implementation as additional therapy to current approaches (Table 1, Table 2 and Table 3).

**Table 1 cells-11-02663-t001:** Human trials of therapeutic strategies via OS in CAVD.

Study and Refs.	Compound	Administration and Doses/Researched Cells	Salient Findings
**Clinical Trials**			
ASTRONOMER [101]	Rosuvastatin vs. placebo	40 mg/day	-Lp(a) and OxPL-apoB levels are associated with faster AS progression; -OxPL-apoB levels were higher after one year in the rosuvastatin arm.
RAAVE [102]	Rosuvastatin vs. placebo	20 mg/day	-Precocious statin treatment is more effective in the progression of aortic valve stenosis.
SALTIRE [103]	Atorvastatin vs. placebo	80 mg/day	-Intensive lipid-lowering therapy delays the progression of calcific aortic stenosis.
SEAS [104]	Simvastatin + Ezetimibe vs. placebo	40 mg + 10 mg/day	-No reduction in valvular or ischemic events in patients with aortic stenosis.
FOURIER [115,117]	Evolocumab	140 mg every 2 weeks or 420 mg every month	-After 1-year of reduced LDL cholesterol levels and cardiovascular events; -Higher Lp(a) levels were associated with a higher risk of AS events.
GLAGOV [118]	Evolocumab	420 mg every month	-Added statin treatment in angiographic coronary artery disease decreased atheroma volume.

Subcutaneously (sq); human valve interstitial cells (hVICs); human umbilical vein endothelial cells (HUVECs); diabetic human aortic endothelial cells (D-HAEC); Hengshun aromatic vinegar (HSAV); malondialdehyde (MDA); glutathione peroxidase (GSH-Px); protein kinase C zeta (PKCζ); homocysteine (Hcy); endothelin 1 (ET-1).

## 4. Conclusions

Aortic stenosis is a significant worldwide medical issue and a disabling disease that diminishes a patient’s life quality. Even though the pathogenesis of aortic stenosis cannot be explained by a single mechanism, current data reveal the deleterious effects of increased OS activity in the complex heterogeneity of active cellular and molecular mechanisms that support the progression of CAVD and endothelial dysfunction through various pathways. Nonetheless, research continues to address new molecules that, in addition to playing roles in cardiovascular pathophysiology, may also represent novel therapeutic targets. Due in part to the fact that the multifactorial, self-replicating cellular mechanisms causing CAVD have already been established by the time a patient presents with the disease, proposed strategies to inhibit oxidative stress, either systemically or locally in CAVD, have so far delivered unclear results.

Furthermore, the spectrum of early paraclinical diagnosis might be expanded; the risk stratification and long-term prognosis could be improved through the development of new protein assays and technologies that speed up and increase the detectability of aortic stenosis. Currently, there are few in vivo/in vitro and cohort observational studies that could determine whether there is a relationship between the degree of valvular oxidative stress, the rate of disease progression, and its relationship to clinical severity. Therefore, more research studies with adequate follow-up periods are essential for translating the OS concept into clinical practice therapies.

## Figures and Tables

**Figure 1 cells-11-02663-f001:**
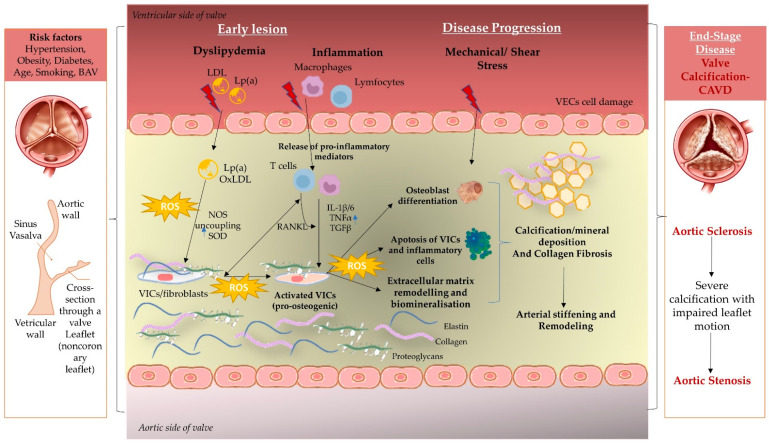
Contribution of oxidative stress in calcific aortic valve disease. Calcific aortic valve disease (CAVD); the bicuspid aortic valve (BAV); reactive oxygen species (ROS); low-density lipoproteins (LDL); lipoprotein (a) (Lp(a)); vascular endothelial cells (VECs); valvular interstitial cells (VICs); nitric oxide synthases (NOS); superoxide dismutase (SOD); receptor activator of NF- kB ligand superfamily member 11 (RANKL); transforming growth factor beta (TGFβ); tumor necrosis factor-alpha (TNF-α).

**Table 2 cells-11-02663-t002:** Human experimental research of therapeutic strategies via OS in CAVD.

Refs.	Compound	Administration and Doses/Researched Cells	Salient Findings
**Experimental Human Studies**			
[120]	IONIS-APO(a)Rx vs. placebo	100 mg, 200 mg, and then 300 mg once a week for 4 weeks each /1 dose of 10–120 mg sq /multiple doses of 10 mg, 20 mg, or 40 mg sq	-Reduced Lp(a) levels in a dose-dependent manner.
[137]	Curcumin	hVICs	-Inhibition of NF-κB, AKT, ERK.
[141]	Cardamonin	In Vitro: hVICs Ex Vivo: human aortic valve leaflet	-Inhibition of VIC osteogenic differentiation through the NF-κB/NLRP3 inflammasome pathway.
[145]	Caffeic Acid	hVICs	-Inhibition of the ERK/AKT/NF-κB/NLRP3 inflammasome pathway.
[152]	Quercetin	HUVECs	-Attenuated atherosclerotic inflammation and adhesion molecule expression by the TLR-NF-κB pathway.
[26]	Anthocyanins	D-HAEC	-Inhibition of the NF-κB pathway.
[169]	Andrographolide	hVICs (from patients undergoing Bentall surgery due to acute type I aortic dissection)	-Inhibition of the NF-κB/Akt/ERK pathway.
[176]	HSAV	HUVECs	-Inhibited apoptosis, decreased serum Hcy, ET-1), ox-LDL levels, MDA level; -Increased NO, eNOS, SOD, GSH, and GSH-Px levels; -Downregulated the expression of PKCζ and regulated the SIRT1-mediated pathway.
[178]	L-carnitine	Patients with RVHD with CPB-induced MIRI (myocardial ischemia-reperfusion injury)	-Increased levels of SOD, CAT; -Suppressed activation of NF-κB and Nrf2.
[24]	Adenoviral SOD delivery	hVIC	-Reduced VIC osteoblastic differentiation by reducing RUNX2, MSX2, and OPN.
[23,181]	CNPs	hVIC	-Scavenged ROS, acted as SOD-mimetics, and reduced VIC osteoblastic differentiation.

Subcutaneously (sq); human valve interstitial cells (hVICs); human umbilical vein endothelial cells (HUVECs); diabetic human aortic endothelial cells (D-HAEC); Hengshun aromatic vinegar (HSAV); malondialdehyde (MDA); glutathione peroxidase (GSH-Px); protein kinase C zeta (PKCζ); homocysteine (Hcy); endothelin 1 (ET-1)**.**

**Table 3 cells-11-02663-t003:** Animal experimental studies of antioxidant compounds and other therapeutic options that target OS with potential in CAVD treatment.

Ref.	Compound	Species and/or Cells Researched	Meaningful Findings
[121]	E06 natural antibody	E06-scFv transgenic mice	-Counteracted the proinflammatory and proatherogenic OxPL effects.
[126]	Resveratrol	Ovariectomized rats	-Reduced RUNX2, ALP expression, and aortic calcification.
[127]	Resveratrol	Rat vascular smooth muscle cells (RASMCs)	-Prevents vascular calcification and mitochondria dysfunction through SIRT1 and Nrf2.
[128]	Resveratrol	Mouse model of uremia	-Fewer aortic atherosclerotic lesions at the site of the ascending aorta.
[131]	Resveratrol	Porcine aortic valve interstitial cells (pVICs)	-Inhibition of osteogenic pVIC differentiation through the AKT/SMAD1/5/8 signaling pathway.
[129]	PLD the natural precursor of resveratrol	Mice with complete ligatures of the left carotid arteries for 14 days	-Reduced adhesion molecule expression (ICAM-1, VCAM-1), proinflammatory cytokine production (TNF-α, IL-1β), iNOS, NF-κB expression, and BAX, Fas-Ligand activation.
[136]	Curcumin	Apolipoprotein E-knockout mice	-Reduced TLR4 expression, macrophage infiltration in atherosclerotic plaque, aortic IL-1β, TNF-α, VCAM-1, ICAM-1 expression, NF-κB activity, and plasma IL-1β, TNF-α, soluble VCAM-1, and ICAM-1 levels; -Reduced the extent of atherosclerotic lesions and inhibited atherosclerosis development.
[138]	Curcumin	Different types of mice, all treated with HF and mice fed with a normal chow diet	-Reduced serum lipid levels, TNF-α, IL-1β, and the aortic atherosclerotic lesion area.
[141]	Cardamonin	In Vivo: mice model fed with a HF diet	-Inhibition of VIC osteogenic differentiation through the NF-κB/NLRP3 inflammasome pathway.
[143]	Ellagic acid	Rat model	-Improved nitric oxide bioavailability and reduced ROS formation.
[144]	Gallic acid	Vascular smooth muscle cell	-Inhibition of vascular calcification through the BMP2-SMAD1/5/8 signaling pathway.
[149]	Nobiletin	Male Wistar rats	-Increased intracellular cGMP (activation of cGC, opening BK channels and KATP channels).
[151]	Quercetin	Adenine-induced chronic renal failure rats	-Modulation of vascular calcification through the iNOS/p38 MAPK pathway.
[154]	Anthocyanins	Tac-induced myocardial dysfunction in mice	-Ameliorated Tac-induced myocardial dysfunction, oxidative stress, and apoptosis via the DDAH1/ADMA/no pathway.
[155]	Puerarin	In Vitro; rat vascular smooth muscle cells In Vivo; uremic rats	-Modulated NLRP3/CASPASE1/IL-1β, NF-κB, and ER/PI3K-AK signaling pathways; -Prevents calcium deposition and inhibits the expression of RUNX2 and BMP2.
[158]	Puerarin	VSMCs	-Inhibited oxLDL-induced VSMC viability via inhibition of the p38 MAPK and JNK signaling pathways; -Decreased the levels of IL-6 and TNF-α and increased SOD activity.
[161]	Diosgenin	Adenine-induced chronic renal failure rats	-Inhibited the c/Akt/ERK, p38 pathway.
[162]	10-DHGD	HCD-fed rabbits	-Alleviated calcium deposition via the downregulation of the BMP2/Wnt3a pathway, OPG/RANK modulation, and raised HDL-C levels.
[164]	Vitamin E	Uremic obese rats	-Prevents osteoblastic differentiation in VSMC and inhibits dephosphorylation of Akt.
[87]	Fucoxanthin	In Vitro; rat heart VIC In Vivo; dog model	-Inhibition of the Akt/ERK pathway.
[172]	Apocynin	VSMCs	-Enhanced expression of α-SMA, reduced expression of BMP2, RUNX2, OPN, suppressed the ERK1/2 pathway and phosphorylation of p47phox (cytosolic NOX2 component).
[70]	Celastrol	In Vitro; porcine AVIC In Vivo; rabbit CAVD model	-Inhibition of NADPH Oxidase 2 and the GSK3β/β-catenin pathway
[174]	Celastrol	Macrophages in mice	-Attenuated oxLDL-induced excessive expression of LOX-1; -Decreased IkB phosphorylation and degradation, reduced production of iNOS, NO, TNF-α, and IL-6; -Reduced atherosclerotic plaque size.
[75]	Glycine	Streptozotocin-induced diabetic rats and HUVECs	-Downregulating the AGE/RAGE signaling pathway by decreasing levels of AGEs, RAGE, NOX4, and NF-κB p65, and by restoring GLO1 function.
[182]	MnBuOE	hVIC and murine model of aortic valve sclerosis	-Inhibited aortic valve remodeling and α-SMA upregulation via TGF-β1; -Upregulated MnSOD via activation of Nrf2.
[52,85]	Mitoquinone	Male Sprague–Dawley rats and adult C57BL/6J mice	-Reduced vascular calcification through the Nrf2/Keap1 pathway and fibrosis by inhibiting the TGF-β1-NOX4-ROS axis.
[184]	DETA NONOate	PAVEC and aortic VIC PAVIC	-Inhibited VIC osteogenic differentiation and calcification.
[186]	L-arginine	Bovine aortic VICs	-Inhibited VIC osteogenic differentiation and remodeling by downregulating ADAMTSL4 and fibrillin-1.
[89]	Anagliptin	Eight-week-old male BALB/c mice	-Activated the PI3K/Akt signaling pathway; -Downregulated the expression of MCP-1, ICAM-1, VCAM-1; -Reduced proteolysis via MMP-2/-9 and CatS/K.
[122,190]	Sitagliptin	Weaned male low-density lipoprotein receptor knockout mice	-Blocked NADPH activation; -Inhibited calcification by downregulating RAGE expression and NF-κB activation.
[189]	Sitagliptin	Rabbit model of CAVD fed with HCD and vitamin D2	-Reduced osteogenic transformation of VICs by reinstating IGF-1 activity.
[198]	Evogliptin	hVIC, endothelial nitric oxide synthase-deficient, male New Zealand white rabbits	-Reduced TNF-α, IL-1β, and IL-6 levels; -Reduced RUNX2 expression.
[69]	Pioglitazone	Mice fed a western-type diet	-Attenuated cusp mobility and inhibited valve calcification by reducing TNFα, IL-6, and BMP2.
[193]	Pioglitazone	Male New Zealand rabbits	-Reduced RAGE activation and inhibited NF-κB p65 intranuclear translocation.

High-fat diet (HF); transverse aortic constriction (Tac); vascular smooth muscle cells (VSMCs); 10-dehydrogingerdione (10-DHGD); high cholesterol diet (HCD); MnTnBuOE-2-PyP5+ (MnBuOE); NO donors application (DETA-NONOate); Porcine aortic VEC (PAVEC); aortic VIC (PAVIC); Polydatin (PLD)**.**

## Data Availability

Not applicable.

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
