# Peer review of "Contribution of Oxidative Stress (OS) in Calcific Aortic Valve Disease (CAVD): From Pathophysiology to Therapeutic Targets"

_cells, 2022, doi:10.3390/cells11172663_

Round 1
Reviewer 1 Report
A very nicely written review on a very topical subject. I particularly like that most of the literature on this difficult topic is covered in a very nice, clear and concise way.
Author Response
Dear Reviewer,
Firstly, we want to thank you in behalf of our team for taken your valuable time to peer-review our manuscript. We are extremely grateful and appreciative for your comment.
Sincerely yours,
Daniela Maria Tanase and her collaborators

Reviewer 2 Report
The manuscript of Dr. Tanase and collaborators is a comprehensive review of the contribution of oxidative stress in calcific aortic valve disease (CAVD). The authors evaluate and describe several mechanisms by which oxidative stress and ROS production / antioxidant systems deficiency contribute to CAVD. The authors also reviewed therapeutical strategies, both experimental and that reached the clinical trial step, somehow related to the management of oxidative stress consequences. The manuscript is well written and covers most of the concepts needed in this field of research and it appropriately considered limitations and pitfalls in the analysis. Nevertheless, there are some minor issues that authors should solve to consider their manuscript for publication in Cells journal.
1) The ‘western diet’ concept (mentioned in row 141) needs to be at least better defined and it would deserve a little more information.
2) Rows 150-151. It would be preferable to report and cite previous work in which the precise definition of Oxidative stress has been provided, such as PMID:32927924.
3) The relationship, if any, between statins’ treatment and oxidative stress that the authors seem to consider in their statement in rows 158-159 has to be explained and clarified. Furthermore, in the 3.1 paragraph of their manuscript, the authors seem to refer to statin treatment as a treatment targeting the oxLDL. Statins can be considered as an LDL-cholesterol lowering drug approach: this approach could have, as a secondary effect, the reduction of oxLDL related to the LDL lowering, but it is hard to consider oxLDL as a primary target of statins. Authors should reformulate their statements or better clarify and discuss their concepts.
4) Rows 335-339. The drugs reported in the studies cited by the authors are designed to inhibit the expression of Apolipoprotein(a) and, thus reduce systemic levels of Lp(a). For the sake of clarity and to avoid misunderstanding, the authors are suggested to better formulate their statement.
5) Table 1. In vitro studies, even if they were performed on human cells, provide results that are still far away from the description of the same phenomenon in humans. Putting clinical trials and in vitro studies on the same level, as it appears to be in the Table presented by the authors, would reduce and negatively impact the key message of the authors. The authors are then suggested to split Table 1 into independent and separated tables.
Author Response
Dear Reviewer,
Firstly, thank you on behalf of our team for your valuable time on peer-reviewing our manuscript. We believe that with the help of your comments we improved the quality of this paper. Thus, we made accordingly the fallowing changes:
1) The ‘western diet’ concept (mentioned in row 141) needs to be at least better defined and it would deserve a little more information.
Response: In section “2.2. Lipid deposition and oxidative stress in SA and BAV”, we have completed the ‘western diet’ concept as suggested and added the corresponding references.
- López-Gil, J.F.; Tárraga-López, P.J. Research on Diet and Human Health. Int. J. Environ. Res. Public. Health 2022, 19, 6526, doi:10.3390/ijerph19116526.
- Afshin, A.; Sur, P.J.; Fay, K.A.; Cornaby, L.; Ferrara, G.; Salama, J.S.; Mullany, E.C.; Abate, K.H.; Abbafati, C.; Abebe, Z.; et al. Health Effects of Dietary Risks in 195 Countries, 1990–2017: A Systematic Analysis for the Global Burden of Disease Study 2017. The Lancet 2019, 393, 1958–1972, doi:10.1016/S0140-6736(19)30041-8.
2) Rows 150-151. It would be preferable to report and cite previous work in which the precise definition of Oxidative stress has been provided, such as PMID:32927924.
Response: Thank you for this observation, we have introduced the reference by which is stated the precise definition of Oxidative stress (line 155-157) corresponding to reference [46], in the main text.
- Sies, H. Oxidative Stress: Concept and Some Practical Aspects. Antioxidants 2020, 9, 852, doi:10.3390/antiox9090852.
3) The relationship, if any, between statins’ treatment and oxidative stress that the authors seem to consider in their statement in rows 158-159 has to be explained and clarified. Furthermore, in the 3.1 paragraph of their manuscript, the authors seem to refer to statin treatment as a treatment targeting the oxLDL. Statins can be considered as an LDL-cholesterol lowering drug approach: this approach could have, as a secondary effect, the reduction of oxLDL related to the LDL lowering, but it is hard to consider oxLDL as a primary target of statins. Authors should reformulate their statements or better clarify and discuss their concepts.
Response:
Indeed, this aspect was unclear, therefore we have rephrased these paragraphs for a better understanding; Lines 164-167: “Although oxidative stress and atherosclerosis are prominent traits in CAVD, statins used as successful monotherapy in atherosclerosis, have not proven to be effective in slowing CAVD treatment amidst the significant evidence that LDL contributes to disease progression.” , and in “3.1 Targeting lipid oxidation: ”Since atherosclerosis and CAVD share similar risk factors and pathways in the initial pathological stages, targeting LDL to further prevent oxLDL formation was the first subject of interest in regard to medical therapies for aortic stenosis”
4) Rows 335-339. The drugs reported in the studies cited by the authors are designed to inhibit the expression of Apolipoprotein(a) and, thus reduce systemic levels of Lp(a). For the sake of clarity and to avoid misunderstanding, the authors are suggested to better formulate their statement.
Response: Line 345-348, we have reformulated this statement and completed as suggested, for better clarity of the information described.
5) Table 1. In vitro studies, even if they were performed on human cells, provide results that are still far away from the description of the same phenomenon in humans. Putting clinical trials and in vitro studies on the same level, as it appears to be in the Table presented by the authors, would reduce and negatively impact the key message of the authors. The authors are then suggested to split Table 1 into independent and separated tables
Response: Following your valuable comments, we have separated Table 1 in two tables: Table 1 containing clinical trials and Table 2 with in vitro studies. We additionally revised the whole manuscript, identified and corrected any other mistakes and adjusted the bibliography.
Sincerely yours,
Daniela Maria Tanase and her collaborators